# Focus-Then-Contact: Speeding Up Robotic Contact-Rich Task Learning with Affordance-Guided Real-World Residual Reinforcement Learning

Guanren Qiao [1]  Ruixiang Ouyang [1]  Sheng Xu [1]  Ruixing Jin [1]  Yueci Deng [2]  Yunxin Tai [2]  Kui Jia [1,2]  Guiliang Liu[†][1]

## Abstract

Real-World Reinforcement Learning (RL) has shown significant potential in robotic manipulation tasks. However, many methods still require substantial human-in-the-loop involvement to complete contact-rich tasks, especially when there are disruptions such as visual backgrounds or positional changes. To address this, we propose the Focus Then Contact (FTC), a lightweight and low-cost method to accelerate the convergence of human-in-the-loop real-world RL for contact-rich tasks. FTC leverages residual RL to provide base actions, helping the system quickly reach the target regions and improve sample efficiency. Additionally, FTC integrates an affordance-guided reward that drives the real-world RL system to quickly focus on key regions of interest, making it possible for the robotic arm to continuously engage with these goal areas through force-control feedback. At the same time, we optimize the human-in-the-loop implementation to prevent conflicts with RL over control of the robotic arm. We demonstrate the effectiveness of FTC on 6 contact-rich tasks, where it outperforms baseline methods in achieving high success rates and speeds up robotic contact-rich task learning under a real-world RL setting. Our website can be seen in https://edem-ai.github.io/FTC-website/.

## 1. Introduction

Reinforcement Learning (RL) has been gradually applied to various robotic tasks in locomotion (Lee et al., 2020;

Margolis et al., 2024; Radosavovic et al., 2024; Qiao et al., 2025; Lin et al., 2025) and manipulation (Chen et al., 2022; Luo et al., 2024; Lu et al., 2025a), enabling robots to autonomously master a range of skills, from simple movements to complex object manipulation. The success of traditional RL is largely based on tasks in simulated environments (Qiao et al., 2024; Rudin et al., 2021; Mittal et al., 2025), where the training process can quickly iterate through numerous failures. However, many contact-rich tasks, such as USB insertion, require high precision and place extremely high demands on simulation engines (Chen et al., 2024; Ankile et al., 2025b; Liu et al., 2025).

Human-in-the-loop real-world reinforcement learning has enabled the transformation of physical environments into practical, interactive training platforms for RL (Luo et al., 2024; 2025). This approach removes the traditional reliance on simulation engines, and the intervention of human demonstrations in real-world RL training can even raise the success rate of contact-rich tasks to 100% (Chen et al., 2025; Stranghöner et al., 2025; Luo et al., 2025). However, due to the limited number of interactions feasible in the real world, the human-in-the-loop approach requires significant time investment from human experts to guide the robotic arm through tasks. Thus, the core challenge of human-in-the-loop real-world RL lies in: **how can we efficiently learn robust and precise control policies within extremely limited physical interactions?**

A promising way to enhance RL policies is by combining them with imitation learning (IL) (Yuan et al., 2025b; Chu et al., 2025). IL provides a base action that regularizes exploration, while online RL improves policy performance through interaction with the environment. However, modern IL models such as Vison-Language-Action (VLA) have complex architectures, making direct application of RL methods challenging (Li et al., 2025a; Lu et al., 2025a; Zang et al., 2025). A natural thought is that Residual RL offers a simpler, yet effective solution by addressing these issues. The core concept involves fixing a "base policy" derived from IL and training a lightweight "residual policy" to iteratively adjust the base policy's outputs in real-time (Alakuijala et al., 2021; Ankile et al., 2025a). However, this

---

[1]School of Data Science, Chinese University of Hong Kong (Shenzhen), Guangdong province, China [2]DexForce Co., Ltd., Guangdong province, China. Correspondence to: Guiliang Liu <liuguiliang@cuhk.edu.cn>.

*Proceedings of the 43rd International Conference on Machine Learning*, Seoul, South Korea. PMLR 306, 2026. Copyright 2026 by the author(s).

approach has yet to fully demonstrate its potential in human-in-the-loop real-world RL settings. Beyond this, many current approaches rely on sparse reward, which leads to low exploration efficiency (Xu et al., 2024; Dong et al., 2025; Ankile et al., 2025a). We need to design a dense reward (Li et al., 2026b) that guides the agent to gradually focus on the region of interest (ROI) that facilitates task completion. One promising direction is to incorporate affordance (the actionable properties of objects in an environment), which can guide the agent's attention toward ROI (Bahl et al., 2023; Lee et al., 2025). However, current models for predicting or extracting affordance (Achiam et al., 2023; Carion et al., 2025) are typically large, which can hinder inference speed in real-world RL.

This paper proposes Focus-Then-Contact (FTC), a lightweight and low-cost human-in-the-loop real-world reinforcement learning framework designed for contact-rich, fine-grained manipulation tasks. FTC advances the state of the art along three key dimensions: (1) *Base action from residual RL*: We leverage residual RL to refine the IL base policy, which provides a base action. It provides a starting point for exploration, allowing the system to focus on the goal region and reduce the complexity of learning in real-world environments. (2) *Keyframe-based Affordance Guidance*: To minimize meaningless contact-based trial and error, FTC provides continuous learning signals derived from keyframe-based affordance. We extract a goal keyframe from the demonstrations and employ a temporally consistent visual encoder to measure the distance between the current and goal embeddings. (3) *Contact through Human-in-the-loop in Cluttered Scenes*: To facilitate precise interaction, FTC employs a dual wrist-camera setup and adds rotational generalization of both the gripper and target objects in cluttered environments. Human experts can intervene through a timed intervention window.

We design various experiments to verify the performance of FTC. Experimental results show that FTC exhibits efficiency, generalization, and a certain of robustness. It can generalize to any position while reducing the human burden during training. We demonstrate that incorporating base policy and an affordance-guided reward does not conflict; instead, they complement each other. FTC provides the community with a lightweight and practical framework for learning contact-rich atomic skills.

## 2. Related Work

**Human-in-the-loop Real-world RL Systems** Recent works successfully integrated human intervention into policy training (Kelly et al., 2019; Retzlaff et al., 2024), and leveraged RL methods to optimize policy performance, enabling precise and dexterous robot manipulation, such as SERL and HIL-SERL (Luo et al., 2024; 2025). Leveraging the advantages of human guidance, recent works greatly improved the robot's performance in dexterous manipulation tasks. HitL integrated human assistance by evaluating the uncertainty of diffusion policies (He et al., 2025), while Rac directly learned policy recovery from human corrections for long-horizon tasks (Hu et al., 2025). Based on a human-in-the-loop RL system, recent works also made progress in fine-tuning VLA models (Xu et al., 2024; Lu et al., 2025b; Kawaharazuka et al., 2025). ConRFT combined both behavior cloning and Q-learning to achieve high sample efficiency (Chen et al., 2025), while HAPO turned to align the reference of actions using human interventions, helping VLA models to avoid unsatisfied motions (Xia et al., 2025). However, these works require prolonged human intervention and substantial computational resources to complete.

**Residual RL Systems** Instead of finetuning the whole policy network, residual RL systems focus to learn corrections using a light-weight network, providing a safety guarantee and sample efficiency (Alakuijala et al., 2021; Silver et al., 2018; Johannink et al., 2019). Recent works made much progress in robot manipulation tasks utilizing residual RL. DP-RRL and EXPO applied residual RL on flow policies, preserving the expressivity of flow policies (Dong et al., 2025; Li et al., 2025b). ResFiT used a residual RL system to finish high degree-of-freedom (DoFs) tasks (Ankile et al., 2025a). CR-DAgger integrated human-in-the-loop with IL and learned from human corrections (Xu et al., 2025). Nearly all recent works utilize sparse rewards for RL training, where rewards are only provided when the task is near completion or successfully finished, leading to extensive ineffective exploration and slow training convergence.

**Affordance for robot manipulation** Affordance is widely used as a way of spatial representation for robot manipulation (Manuelli et al., 2019; Ardón et al., 2020; Gao & Tedrake, 2021; Curtis et al., 2022; Xu et al., 2026). By predicting different representations of affordance, prior works have achieved success in visual-language models (VLM) for real-world robot manipulation (Bahl et al., 2023; Yuan et al., 2025a; Liu et al., 2024). As a useful guidance for object interactions, recent works also integrated an RL system with affordance signals (Brohan et al., 2023; Fang et al., 2023). KAGI used a VLM to generate affordance from robot observations, and then computed dense rewards to guide RL training (Lee et al., 2025). However, the high computational cost of VLM inference can not satisfy the real-world RL training. Thus, our work leveraged a more lightweight way to provide affordance-guided rewards for real-world RL training.

# 3. Problem Formulation

**Goal-Conditioned Partially Observable Markov Decision Process.** We formulate the real-world RL robotic contact-rich manipulation tasks as a infinite-horizon Goal-Conditioned Partially Observable Markov Decision Process (GCPOMDP) (Sutton & Barto, 2018) defined by the tuple $\mathcal{M} = (\mathcal{O}, \mathcal{S}, \mathcal{A}, P_\mathcal{T}, \mathcal{R}, \mu_0, \gamma, g, \phi)$, where: 1) Within the observation space $\mathcal{O}$, each observation $o_t \in \mathcal{O}$ consists of two components: proprioception ($o_t^p$) and visual input ($o_t^v$). The proprioception $o_t^p$ includes the 6D pose of the end-effector, along with its linear velocity, angular velocity, and the 6-dimensional force-torque vector. Meanwhile, the $o_t^v$ represents the RGB images captured by the wrist cameras located on both sides of the end-effector. 2) $s_t \in \mathcal{S}$ records the complete information of the robot, workspace environment, and surrounding background. We summarize a state as $s_t = [o_t, o_{t-1}, ..., o_{t-H}]$ and each $o_t = [o_t^p, o_t^v]$. 3) $\mathcal{A}$ denotes the action space, and action $a \in \mathcal{A}$ denotes the 6D pose delta from the last frame of the end-effector, and a one-dimensional discrete value 0/1 is used to control the gripper on/off. 4) $r_t = \mathcal{R}(o_t^v, a_t, g)$ denotes the reward functions, which typically consist of affordance-guided rewards and sparse task rewards. 5) $P_\mathcal{T} \in \Delta_{\mathcal{S} \times \mathcal{A}}^\mathcal{S}$ denotes the transition function as a mapping from state-action pairs to a distribution of future states. 6) $\mu_0 \in \Delta^\mathcal{S}$ denotes the initial state distribution. 7) $\gamma \in (0, 1]$ denotes the discounting factor. 8) $g \in \mathcal{G}$ denotes the target goal (different 6D pose of the end-effector) reached by the robotic arm after completing the task. 9) $\phi(o_t^v) \in \mathcal{G}$ denotes a function that maps a visual observation from the state space to a goal $g$ within the target space. Under this GCPOMDP, our goal is learning a control policy $\pi : \mathcal{S} \to \mathcal{A}$ that can maximize the discounted cumulative rewards $\sum_{t=0}^{T-1} \gamma^t r_t$. The state-action value function can be estimated as:

$$
\begin{aligned}
Q^\pi(o_t, a_t) &= \mathbb{E}_{\mu_0, o_t \sim \mathcal{M}, a_t \sim \pi} \left[ \sum_{t=0}^{T-1} \gamma^t r_t \mid o_t, a_t \right] \\
&= \mathbb{E}_{\mu_0, o_{t+1} \sim \mathcal{M}} \left[ r_t + \gamma \mathbb{E}_{a_{t+1} \sim \pi} [Q^\pi(o_{t+1}, a_{t+1})] \right]
\end{aligned}
\tag{1}
$$

where $T$ represents the task horizon. Our method builds on an off-policy RL algorithm Soft Actor-Critic (SAC) (Haarnoja et al., 2018). We want to solve:

$$
\pi^* = \arg\max_\pi \mathbb{E}_\pi \left[ \sum_{t=0}^{T-1} r_t + \alpha \mathcal{H}(\pi(\cdot|o_t)) \right]
\tag{2}
$$

where $\alpha$ is a regularization coefficient that controls the importance of the entropy term. In line with the human-in-the-loop approach employed in HIL-SERL, human experts can intervene at any time during RL training. The action mode switches to $a_t = \pi_{RL}(o_t)|\pi_{human}(o_t)$ according to an intervention $l$. Such interventions are stored separately into the RL replay buffer $\mathcal{D}_{online}$ and the expert demonstration buffer $\mathcal{D}_{offline}$.

# 4. First "Focus" Then "Contact"

Our method Focus-Then-Contact (FTC) consists of an offline imitation learning pre-training phase and an online RL fine-tuning phase. It is composed of three major components: the residual RL design, the affordance-guided reward, and the human-in-the-loop real-world RL system, as illustrated in Figure 1. In this section, we provide a detailed description of the methods used in the paper.

## 4.1. Real-World Residual Reinforcement Learning

The BC process produces a base policy $\pi_{base}$, which provides a certain level of task success but still leaves room for improvement. To address this, we freeze the $\pi_{base}$ and introduce a residual policy $\pi_{res}$ to correct the mistakes made by $\pi_{base}$ and improve overall performance. The residual network for learning the residual policy is very lightweight, consisting of only two layers of MLP and layer normalization. This refinement ensures two critical aspects: (1) $\pi_{res}$ is loosely coupled with the $\pi_{base}$, enabling seamless integration of different base policies; (2) $\pi_{base}$ helps RL reach the target region more quickly, thereby reducing exploration.

In the residual learning framework, we decompose the action $a_t$ into two components: the base action $a_t^{base}$ and the residual action $a_t^{res}$. The $a_t^{base}$ also needs to be stored in the $\mathcal{D}_{offline}$ and $\mathcal{D}_{online}$. Since the action contains three dimensions representing rotation in Euler angles, rotations generally cannot be directly added. However, this does not have a significant impact on the convergence of the neural network. Thus, the total action at time $t$ is $a_t = a_t^{base} + a_t^{res}$. The residual policy $\pi_{res}$ learns how to adjust the base action by refining it to correct mistakes and improve performance. But when humans intervene, the action taken at that moment does not consider the base policy and is still the final output action. To represent the state-action value $Q^\pi(o_t, a_t^{base} + a_t^{res})$ in the residual framework, we should change the Bellman Equation to approximate the Q-value for the sum of the base action and the residual policy's action. Thus, we should compute Q value with

$$
Q^\pi(o_{t+1}, a_{t+1}^{base} + \pi_{res}(\cdot|o_{t+1}, a_{t+1}^{base}))
\tag{3}
$$

And our problem becomes:

$$
\arg\max_{\pi_{res}} \mathbb{E} \left[ \sum_{t=0}^{T-1} r_t + \alpha \mathcal{H}(\pi_{res}(\cdot|o_t, a_t^{base})) \right]
\tag{4}
$$

During interaction with the environment, the base policy outputs relative values for $k$ end-effector poses each time. To reduce the difficulty of RL learning, we don't let the

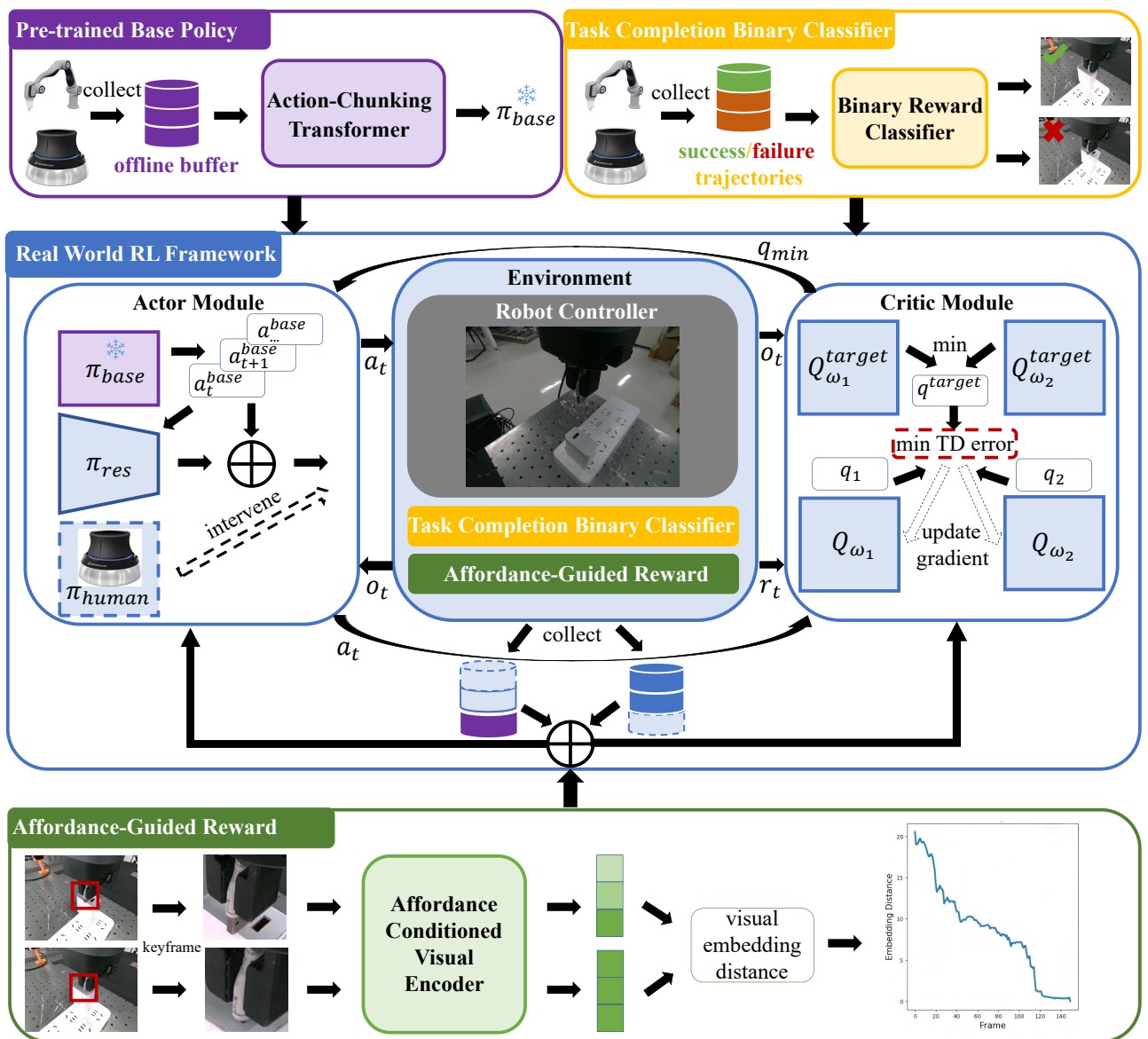

*Figure 1.* **Focus-Then-Contact Framework**: Core idea is that it first focuses on key contact regions and then performs precise contact-rich manipulation. Before real-world RL, a base policy $\pi_{base}$ is trained from offline data, while a task-completion binary classifier is learned from success/failure trajectories to provide sparse rewards. In addition, a dense affordance-guided reward is constructed by encoding the keyframe and computing the distance between the current and the goal embedding. During real-world interaction, the base policy is combined with a SAC-based residual policy $\pi_{res}$ and optional human intervention to enable efficient exploration.

residual policy output all $k$ values. Instead, it only outputs the first value, and the base policy uses this value to generate the final action.

## 4.2. Keyframe-based Affordance-Guided Reward

Although residual RL can improve sample efficiency, sparse rewards still provide limited feedback, which makes it difficult for real-world RL to receive effective guidance during training, resulting in longer training times. RL requires a large number of successful trajectories to learn the correct policy. Additionally, the interaction frequency in real-world RL is much lower than in simulation, demanding a large number of correct examples for the agent to gradually learn and complete the task. This highlights the need for extensive human-in-the-loop teaching to accelerate task learning. In comparison, dense rewards provide continuous feedback that better guides the agent towards success. FTC aims to provide this type of feedback by augmenting sparse task completion rewards with dense process rewards.

Specifically, we think dense rewards are calculated based on

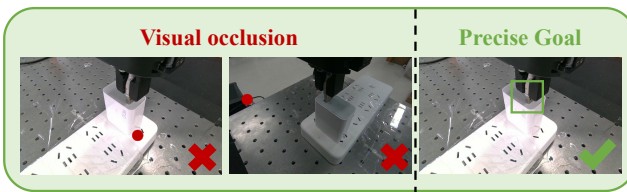

*Figure 2.* Reasons for choosing keyframes: Visual occlusions arising from different camera viewpoints significantly degrade keypoint detection performance, whereas keyframe-based representations indicate these issues.

---

**Algorithm 1** Focus-then-Contact

1: Randomly initialize Critic $Q_i$ (set targets $\tilde{Q}_i = Q_i$) for $i = 1, 2$ and Actor $\pi_{res}$. Choose discount $\gamma$, temperature $\eta$ and critic EMA weight $\rho$, intervention $l$, batch size $N$, delay ratio $c$. Load base policy $\pi_{base}$ and Goal keyframe $g$.
2: Initialize empty replay buffer $\mathcal{D}_{online}$
3: Initialize buffer $\mathcal{D}_{offline}$ with offline data
4: **while** True **do**
5:     Receive initial observation state $o_0$
6:     **for** $t = 0$ to $T$ **do**
7:         Take residual action $a_t^{res} \sim \pi_{res}(\cdot|o_t, a_t^{base})$ and base action $a_t^{base} \sim \pi_{base}(\cdot|o_t)$
8:         $a_t^{RL} = a_t^{base} + a_t^{res}$ or Take human intervention $a_t^H \sim \pi_{human}(\cdot|o_t)$
9:         Get $r_t = \mathcal{R}(o_t, a_t^{RL}/a_t^H, g)$ with Equation 8
10:        Store transition $(o_t, a_t^{RL}/a_t^H, r_t, o_{t+1})$ in $\mathcal{D}_{offline}$ and $\mathcal{D}_{online}$
11:        **for** 1 to $c$ **do**
12:           Sample batch $b_{online}$ of $\frac{N}{2}$ from $\mathcal{D}_{online}$
13:           Sample batch $b_{offline}$ of $\frac{N}{2}$ from $\mathcal{D}_{offline}$
14:           $y_t = r_t + \gamma \min_{i \in \{1,2\}} \tilde{Q}_i(o_{t+1}, a^{base} + a^{res}) + \gamma\eta \log \pi_{res}(\cdot|o_{t+1}, a_t^{base})$ with **b**
15:           Update Q networks:
16:           $L^{critic} = \frac{1}{N}\sum(y_t - \min_{i=\{1,2\}} Q_i(o_t, a_t^{RL}/a_t^H))^2$
17:           Update $\tilde{Q}$ networks with EMA weight $\rho$
18:        **end for**
19:        With **b**, update $\pi_{res}$ maximizing objective:
20:           $\mathbb{E}\left[\min_{i=\{1,2\}} Q_i(o_t, a_t^{base} + a_t^{res})\right.$
21:           $\left. -\eta \log \pi_{res}(\cdot|o_t, a_t^{base})\right]$
22:     **end for**
23: **end while**

---

affordance information. Traditional methods often rely on using keypoints to represent affordances (Lee et al., 2025; Ardón et al., 2020). Initially, we also adopted this approach, specifically using a VLM (Achiam et al., 2023)/object detection + segmentation model such as SAM 3 (Carion et al., 2025) with visual or language prompts to identify keypoints. Although VLM/SAM 3 has strong generalization capabili-

ties, directly applying this method comes with some unresolved issues: (1) The generalization ability of open-source VLMs is insufficient (Li et al., 2026a), but collecting a large amount of data for fine-tuning is too costly. Closed-source VLMs, on the other hand, do not meet the inference speed requirements because real-world RL needs to provide rewards in real-time. (2) Figure 2 illustrates that the object detection in SAM 3 can be affected by the surrounding environment, making it prone to errors in cluttered backgrounds. (3) Our experimental task setup involves rotational generalization. During the gripper rotation process, visual occlusion occurs, affecting the camera view and leading to potential errors in the perception.

We then consider *whether keyframe-based affordance could serve as an alternative to guide RL towards the goal?* For tasks such as USB insertion, the hole is often occluded during the manipulation process, but we only need to focus on the goal state at task completion. Before real-world RL training, the $\mathcal{D}_{offline}$ is pre-filled with demonstration trajectories, from which we can directly extract the key ROI relevant to the task completion. Based on this, we design a dense reward function: both the current image and the goal keyframe are cropped to the same predefined ROI, then encoded by a shared visual encoder. The reward is computed as the distance function $d(\cdot)$ between the two embeddings.

In our experiments, we find that directly using traditional visual encoders (e.g., ResNet or ViT) to compute the distance and obtain reward signals resulted in unstable performance. To obtain smoother and more geometrically consistent embeddings, we refer to (Ma et al., 2023), which learns a self-supervised optimal distance function. This can be formulated in the InfoNCE (Oord et al., 2018) format:

$$\min_d \mathbb{E}_{p(g), p^+(o_t; g)}\left[-\log \frac{e^{d(\psi(o_t); \psi(g))}}{Z}\right] \quad (5)$$

$$Z = \mathbb{E}_{p^-(o_t, o_{t+1}; g)}\left[e^{\left(\delta_g(o_t) + \beta d(\psi(o_{t+1}); \psi(g)) - d(\psi(o_t); \psi(g))\right)}\right] \quad (6)$$

where $p(g)$ can be thought of as the distribution of the goal keyframe, $p^+(o_t; g)$ the distribution of "positive" samples, and $p^-(o_t, o_{t+1}; g)$ the distribution of "negative" samples. $\beta$ is a hyper-parameter. This can be understood as a novel implicit time contrastive objective that generates a temporally smooth embedding, enabling the distance function to be implicitly defined via the embedding distance. Then we construct the reward function for our task:

$$\mathcal{R}_{dense}(o_t^v, a_t^{RL}/a_t^H, g) = -||\psi(o_t^v) - \psi(g)||_2^2/\alpha \quad (7)$$

where $\alpha$ is a hyper-parameter. We also train a sparse binary reward classifier and combine it with $\mathcal{R}_{dense}$:

$$r_{\text{total}} = \begin{cases} 1, & \text{if } r_{\text{sparse}} = 1 \\ r_{\text{dense}}, & \text{if } r_{\text{sparse}} = 0 \end{cases} \quad (8)$$

After experimental verification, we find that such guidance facilitates more efficient and generalizable online learning compared to sparse rewards alone.

### 4.3. FTC Human-in-the-loop Real-World Reinforcement Learning Design

As described above, our approach uses off-policy data, hybrid reward functions, and interleaves using the SAC to train the policy. In the following, we detail the key design choices that are required to achieve a stable FTC framework across different tasks and grippers. Algorithm 1 presents the pseudocode of the FTC framework. For other specific details, please refer to the Appendix B.

**Human-in-the-loop Module Design.** We use an industrial collaborative robot arm, and for the coordinate system, we employ the workpiece coordinate system that is common in industrial applications. In our experiments, we observe occasional conflicts between human demonstrations and RL for control of the robotic arm. To address this issue, we introduce a short intervention window, denoted as $W$, during human demonstrations. If no human demonstration takes place within this window, control is transferred to RL once the window expires. If the window is active but no human demonstration occurs, the robotic arm remains stationary. If the human demonstration persists beyond the window, the window period is reset.

**Reward Scalability.** We discuss dense reward and sparse reward separately. For sparse reward, we train a separate binary reward classifier for each task. We need to collect four successful trajectories and eight failed trajectories per task, with each trajectory containing 50 to 100 images. Training and inference are done on a single 4090 GPU for 2,000 epochs, but the total time does not exceed five minutes. Therefore, even training a sparse reward model individually for each task is very fast.

For dense reward, we use a shared visual encoder for all tasks. To ensure that the visual encoder generalizes to most tasks, it pre-trained on a large amount of human manipulation videos. Although out-of-domain videos are not in-domain data for robot control, they can be considered in-domain data for human action strategies. Figure 3 presents the dense reward curves for various tasks under the same visual encoder, indicating that the keyframe-based reward can transfer across related tasks and objects. It clearly illustrates the fluctuations during different contact-rich tasks.

## 5. Experiment

**Experiment Settings.** To conduct a comprehensive evaluation, we quantify the performance of FTC in 6 contact-rich manipulation tasks from the following perspectives: 1) *Effi-ciency*: Can the FTC significantly reduce the time required for human demonstration to complete tasks? 2) *Robustness*: How well does FTC perform with diverse contact-rich manipulation tasks under position and visual changes? 3) *Generalization*: Can FTC maintain a certain level of success rate in Out-of-distribution (OOD) settings as well?

We use the Diana force-controlled robotic arm, which has 7 degrees of freedom per arm and employs impedance control. However, interaction with the arm is still achieved by sending end-effector pose commands. For teleoperation, we use a 3D SpaceMouse to send incremental end-effector changes to the robotic arm during both human-in-the-loop operation and data collection. The output range is regulated by a scale factor. The cameras we employ are RealSense D405 depth cameras. The two cameras are mounted symmetrically on the wrists of the arms. We choose ACT as our base policy. The codebase of ACT comes from EmbodinChain (Developers, 2025).

**Task Description.** We design 6 contact-rich atomic skill manipulation tasks, including: (1) *Card insertion*: Insert a thin but relatively tall card into a base slot. (2) *Door opening*: Use the robotic arm's gripper to open the door of a storage box. (3) *Charger grasp-insertion*: Grasp a charger and put it into a power strip. (4) *Keychain hanging*: Hang a keychain onto a hook on the storage box. (5) *USB insertion (easy)*: Insert a USB plug into a fixed charger. (6) *USB insertion (hard)*: Insert a USB plug into a charger wherever it is located. The entire experiment is conducted in a natural industrial environment without any black cloth occlusion. Please refer to A for other specific details.

**Metrics.** In our experiments, we adopt three key metrics: (1) *Success Rate*: The ratio of successful task completions within the allotted time to the total number of test trials. (2) *Training Time*: Across all experimental settings, we cap the total human-in-the-loop real-world RL training time at 2 hours. (3) *Cycle Time*: The time required to complete a single task instance from start to success.

**Baselines.** Our method is compared with five other approaches: (1) *BC (MLP)*: Behavioral Cloning using a Multi-Layer Perceptron network, a baseline imitation learning method that directly maps states to actions via supervised learning. (2) *ACT* (Zhao et al., 2023): Action Chunking with Transformers, a method that leverages transformer architecture to generate action sequences in chunks for robotic manipulation. (3) *ResFiT* (Ankile et al., 2025a): A residual RL framework with sparse reward and human supervision for resets. (4) *HG-DAgger* (Kelly et al., 2019): A variant of DAgger that is more suitable for interactive imitation learning from human experts in real-world environments. (5) *HIL-SERL*: The first human-in-the-loop vision-based real-world RL system.

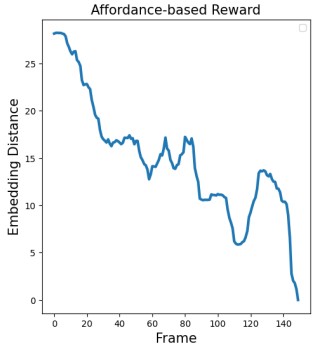 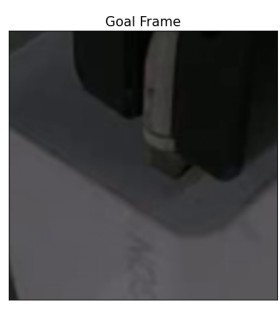 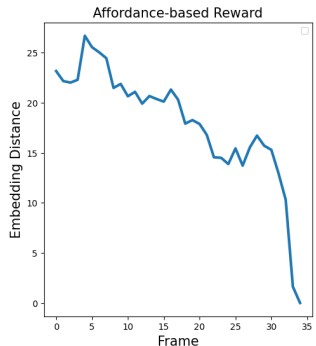 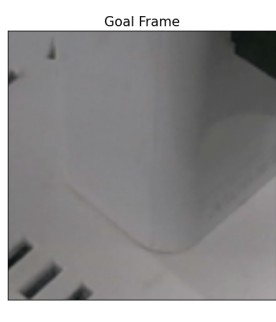

*Figure 3.* The dense reward curves for various tasks.

*Table 1.* Performance Comparison of Baselines Across Different Tasks: We test each method 50 times across different positional variations. To ensure a fair comparison, we train BC (MLP) and ACT using 50-100 data samples. For ResFiT, HG-Dagger, and HIL-SERL, we initially used 40-50 expert trajectories to fill $D_{offline}$ . The $\pi_{base}$ for FTC was pretrained with 40 data samples, while expert demonstrations were initially filled with 20-30 trajectories in $D_{offline}$.

| Baseline | Card insertion | Door opening | Charger grasp-insertion | Keychain hanging | USB insertion (easy) | USB insertion (hard) | Average |
|---|---|---|---|---|---|---|---|
| | | | **Success Rate (%)↑** | | | | |
| BC(MLP) | 0 | 0 | 0 | 0 | 0 | 0 | 0 |
| ACT | 0 | 0 | 8 | 0 | 0 | 0 | 1.3 |
| HG-DAgger | 0 | 84 | 70 | **100** | 26 | 0 | 46.7 |
| ResFiT | 0 | 0 | 0 | 0 | 0 | 0 | 0 |
| HIL-SERL | **100** | **100** | **100** | **100** | 92 | 0 | 82 |
| FTC | **100** | **100** | **100** | **100** | **100** | **100** | **100** |
| | | | **Training Time (min)↓** | | | | |
| HIL-SERL | 60 | 35 | 35 | 30 | 60 | ≥120 | ≥56.7 |
| FTC | **35** (1.71x faster) | **20** (1.75x faster) | **20** (1.75x faster) | **20** (1.5x faster) | **25** (2.4x faster) | **45** (+∞x faster) | **27.5** |
| | | | **Cycle Time (seconds)↓** | | | | |
| HIL-SERL | 22.8 | 17.3 | **13.1** | **15.7** | 11.0 | - | - |
| FTC | **20.9** | 17.1 | 14.8 | 16.8 | 10.6 | **11.5** | **15.3** |

*Table 2.* Ablation Study: We evaluate each method 20 times at each task.

| Baseline | Card insertion | Charger grasp-insertion | USB insertion (hard) | Average |
|---|---|---|---|---|
| | | **Success Rate (%)↑** | | |
| FTC (w/o human-in-the-loop) | 0 | 0 | 0 | 0 |
| FTC (w/o affordance) | 100 | 100 | 100 | 100 |
| FTC (w/o Residual RL) | 100 | 100 | 100 | 100 |
| FTC (w/o Residual RL + affordance) | 100 | 100 | 0 | 66.7 |
| FTC | 100 | 100 | 100 | 100 |
| | | **Training Time (min)↓** | | |
| FTC (w/o human-in-the-loop) | ≥120 | ≥120 | ≥120 | ≥120 |
| FTC (w/o affordance) | 50 | 25 | 74 | 49.7 |
| FTC (w/o Residual RL) | 50 | 33 | 85 | 56 |
| FTC (w/o Residual RL + affordance) | 60 | 35 | ≥120 | ≥71.7 |
| FTC | **35** | **20** | **45** | **33.3** |

## 5.1. Effectiveness: Accelerating the convergence of human-in-the-loop real-world RL

To comprehensively evaluate the performance of the proposed FTC framework in real-world contact-rich fine-grained manipulation tasks, we design a series of tasks covering different complexities and generalization requirements, and conduct a systematic comparison with current mainstream methods. Experimental results demonstrate that FTC significantly outperforms existing methods in terms of success rate and training efficiency.

As shown in Table 1, FTC achieves a 100% success rate across all 6 tasks, significantly outperforming all baseline methods. In contrast, behavior cloning-based methods (BC (MLP) and ACT) and the Residual RL method ResFiT fail in almost all tasks, indicating that relying solely on imitation learning without human-in-the-loop struggles to adapt to contact-rich manipulation. While HG-Dagger and HIL-SERL also incorporate human interaction, their success rates remain substantially lower than FTC (averaging 46.7% and 82%, respectively). In terms of training efficiency, FTC requires only 27.5 minutes on average, more than twice as fast as HIL-SERL (≥56.7 minutes). For the most challenging USB-insertion (hard) task, FTC achieves 100% success in just 45 minutes, whereas HIL-SERL fails to converge within 120 minutes. Regarding execution efficiency, FTC does not suffer from increased inference latency due to the overhead of base-policy reasoning; moreover, it demonstrates the same stability in the hard USB-insertion setting as in the easy setting. We record the number of human intervention steps during training, and Figure 4 shows that in these tasks, FTC generally exhibits a faster and more consistent reduction in human intervention, indicating better

overall performance and quicker adaptation than HIL-SERL. Compared to HIL-SERL, FTC reduces human intervention by more than half to complete these tasks.

To investigate the contribution of each core component of FTC, we conduct a systematic ablation study (See Table 2). After removing the human-in-the-loop module, the model failed to converge on all tasks, highlighting the critical role of human demonstrations in early exploration and error correction. When either the affordance-guided reward or the Residual RL module is removed, the model still achieves a high success rate, but the training time significantly increases (with averages of 49.7 minutes and 56 minutes, respectively). When both Residual RL and affordance-guided reward are removed, the model completely failed on the USB insertion (hard) task. This indicates that residual RL and affordance-guided reward have a synergistic effect in enhancing learning efficiency and task generalization.

Through these experiments, we reflect on *the reasons why FTC is both fast and accurate*. For example, in a plug-in task, it can be divided into two stages: the descent and alignment phases. During the descent phase, the base policy primarily leverages its capability to focus on regions close to the ROI. As it continues to make contact, the affordance-guided reward provides feedback on how far it is from the insertion hole, helping to guide the arm. During the descent phase, affordance-based feedback isn't particularly helpful because the difference between the image embedding and the goal embedding is still quite large. However, as the robotic arm gets closer to making contact, this dense reward becomes crucial in capturing whether the robot arm is approaching the insertion hole. We have included some trends of the dense reward curves on the website. Thus, both the base policy and affordance-guided reward work together in a complementary manner.

### 5.2. Robustness: Evaluating the impact of position and visual disturbances on task performance

Our experimental setting is based on the rotational and translational variations in position, which can be seen in Appendix D. FTC can clearly adapt to position changes in a short period, quickly identifying the ROI to complete the task (See Table 1). To test FTC's robustness under non-ideal visual conditions, we conduct three experiments under challenges such as environmental lighting changes and scene occlusions (Table 3). The experimental results show that, under environmental background changes, the task success rate remained 100%. We think FTC exhibits a certain degree of visual robustness, mainly due to: 1) Affordance guidance: By focusing on ROI directly related to task completion, rather than the global background, FTC reduces sensitivity to irrelevant environmental changes. 2) Independence from third-party cameras: The task can be

completed using only the wrist camera, without relying on a third-person perspective. However, when facing strong lighting conditions and occlusions in the scene, contact-rich tasks such as charger insertion are still significantly impacted. We redefine the scope of robustness. We think the FTC is robust at the motion execution level and against unstructured dynamic disturbances in time and space (e.g., object position shifts, waving hands), but the visual perception layer is sensitive to variations in color, texture, lighting color, and lighting intensity. For the failed tasks, we think the primary cause of failure is the insufficient robustness of the visual encoder. Since the object colors in the training data are largely fixed, the feature maps produced under different colors or strong lighting conditions all affect the FTC. Additionally, different colors and strong lighting introduce errors in ROI localization.

### 5.3. Generalization: Ensuring task performance in OOD positions

To further validate the generalization capability of FTC in unseen scenarios, we design a position generalization experiment based on the task of USB insertion (hard). The training is conducted at positions (Pos 1-4) (See Appendix C), and when tested at four unseen target positions (Pos 5-8), FTC still achieves an average success rate of 86.3% in Table 4, while HIL-SERL fails at all positions. Figure 5 demonstrates the performance of the robot arm in the USB insertion (hard) task. Initially, the robot arm is able to insert the USB steadily within the training distribution. However, when we randomly place it at an OOD position Pos 5, the robot arm is still able to insert the USB successfully. This illustrates that the model is capable of generalizing to unseen positions. After removing residual RL, the generalization performance of FTC significantly drops (with an average success rate of only 11.3%), indicating that learning residual terms is more beneficial for adapting to new environments than directly learning the actual actions.

*Table 3.* Vision Disturbance Robustness Experiment: We try each method 10 times at each condition.

| Task | Environmental Changes | Strong Lighting | In-scene Occlusions |
|---|---|---|---|
| | Success Rate (%)↑ | | |
| Charger grasp-insertion | 100 | 0 | 10 |
| Keychain hanging | 100 | 80 | 0 |
| USB insertion (hard) | 100 | 30 | 0 |

**Limitation** (1) Non-monotonic Visual Changes: Obstacle-avoidance pushing: During the process of pushing an object to a target location, the relative pose between the robot and the object, as well as the object's orientation, continuously changes. Visually, the embedding distance between the current frame and the keyframe of "object at target position" is not monotonically decreasing. To navigate around obstacles or adjust the pushing direction, the policy may need to

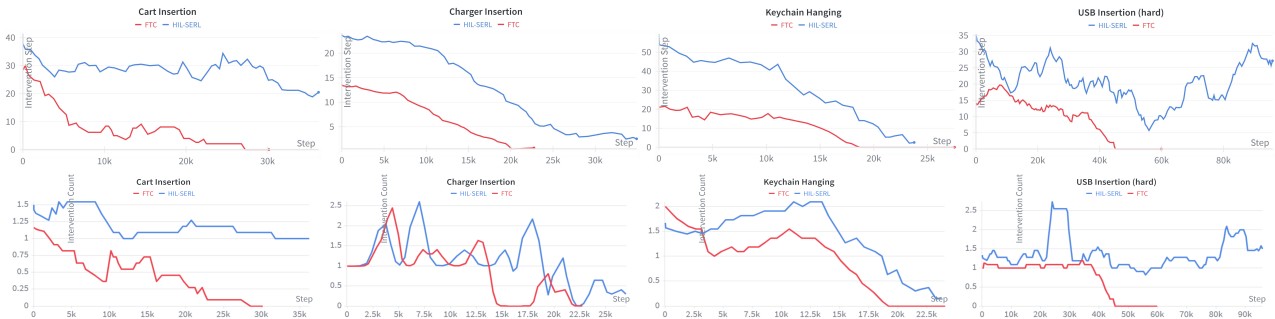

*Figure 4.* Human intervention steps and human intervention counts in four tasks, comparing the performance of HIL-SERL and FTC. The blue line represents HIL-SERL, while the red line represents FTC. The x-axis denotes steps, and the y-axis shows the number of human intervention steps and counts required.

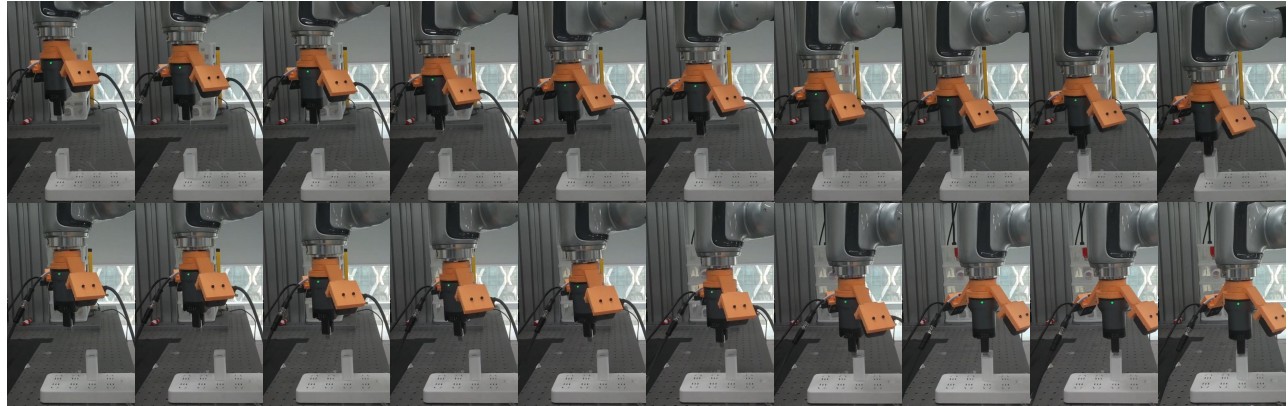

*Figure 5.* OOD example: The first row shows a position within the distribution, and the second row shows an OOD position.

*Table 4.* Out-of-Distribution Generalization Experiment: We test each method 20 times at each position.

| Baseline | Pos 5 | Pos 6 | Pos 7 | Pos 8 |
|---|---|---|---|---|
| Success Rate (%)↑ | | | | |
| HIL-SERL | 0 | 0 | 0 | 0 |
| FTC (w/o Residual-RL) | 15 | 0 | 30 | 0 |
| FTC | **90** | **90** | **95** | **80** |

temporarily move the object away from the target, increasing the visual distance, but this is necessary for the correct strategy. Distance-based rewards would incorrectly penalize such essential behaviors. (2) Process-oriented Tasks: Cloth folding: Successful folding does not depend on reaching a specific "keyframe" configuration of the final cloth state, but rather on executing a correct sequence of operations (e.g., lifting one corner, moving it to the opposite side, placing it down, smoothing). During most intermediate steps, the embedding distance between the current visual state and the final "fully folded" state is large and meaningless. The reward function cannot inform the policy that "lifting this corner" is a good intermediate action, as it does not bring the state any closer to the final keyframe.

## 6. Conclusion

We present Focus-Then-Contact (FTC), a lightweight and low-cost human-in-the-loop reinforcement learning framework for contact-rich, fine-grained real-world manipulation. By combining keyframe-based affordance-guided rewards with residual RL on top of a base policy, FTC achieves fast convergence, generalization to unseen settings, and a certain degree of robustness under real-world disturbances. Our results show that base policies and affordance-guided rewards are complementary rather than conflicting, jointly enabling efficient and stable learning on real robots. We think FTC provides a practical foundation for scalable real-world RL. The base policy, dense reward, and human intervention tools can all be substituted with alternatives. Future work will explore more contact-rich long-horizon manipulation tasks that go beyond atomic skills.

## Impact Statement

This work studies learning-based robotic manipulation for contact-rich tasks that require precise alignment and control, such as insertion, opening, and object placement. The proposed approach aims to improve the robustness and efficiency of robotic systems when interacting with everyday

objects in structured environments.

The potential positive impact of this research includes enabling more reliable automation in settings such as manufacturing, logistics, and assistive robotics, where accurate manipulation can reduce human workload and improve operational safety. By improving success rates and generalization in fine-grained manipulation tasks, this work may contribute to the deployment of robots in real-world applications that require physical interaction.

## Acknowledgments

This work is supported in part by Shenzhen Science and Technology Program under grant KJZD20240903104008012, Shenzhen Science and Technology Program under grant ZDCY20250901113000001, CUHK-CUHK(SZ)-GDSTC Joint Collaboration Fund No. 2025A0505000053, and GuangDong Key Laboratory of Big Data Computing (2021B1212040002).

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

# Supplementary Materials

# A. Task Setup and Policy Training Details

In this section, we provide details regarding how each task is set up, including hardware and software; as well as details on policy training.

## A.1. Card Insertion

In the Card Insertion task, the robot is required to insert a thin upright card into a narrow slot on a fixed base. The card is initially grasped by the robot and positioned above the base with small variations in position and orientation. Successful completion of the task requires precise alignment between the card and the slot, as well as careful control during the insertion phase to avoid collisions with the slot edges. This task emphasizes fine-grained positional accuracy, orientation control, and contact-aware manipulation under tight geometric constraints.

### A.1.1. CROPPED IMAGES

We cropped the images to focus on the task-relevant parts of the scene, as shown in Fig. 6.

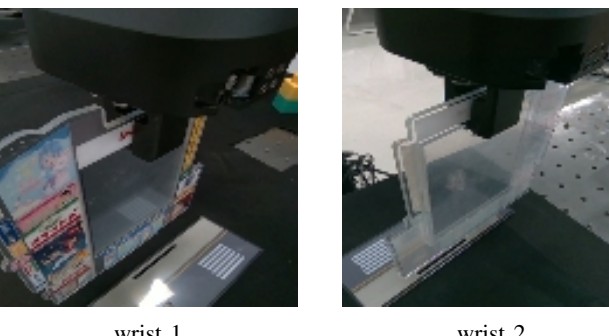

wrist_1        wrist_2

**Figure 6.** Sample input images from cameras used as inputs to the policy.

### A.1.2. POLICY TRAINING DETAILS

In Table 5, we report additional details of the policy training for this task.

**Table 5.** Policy training details for the Card Insertion task.

| Parameter | Value |
|---|---|
| base policy | ACT |
| Observation space | wrist_1, wrist_2, tcp_pose, tcp_vel, tcp_f/t |
| Action space | 6D twist |
| Classifier views | wrist_1 |
| Classifier accuracy | 99% |
| Initial offline demonstrations | 30 |
| Max episode length | 100 environment steps |
| Reset method | Auto reset |
| Randomization range | 10 cm in x and y, 30 deg in rz |
| Proprio encoder size | 128 |
| Residual Policy MLP size | 256×256 |
| Discount factor | 0.98 |
| Optimizer | Adam |
| Image augmentation | Random crop + Random lighting intensity |
| xyz bound | 16cm*15cm*18cm |
| rpy bound | $0*0*[-60, 60]deg$ |

## A.2. Door Opening

In the Door Opening task, the robot uses a parallel-jaw gripper to pull open the top drawer of a plastic container. The gripper first establishes contact with the drawer handle and then applies a pulling motion to slide the drawer along its rails. Successful execution requires reliable contact establishment, appropriate force application, and stable control throughout the opening motion. This task highlights the challenges of contact-rich manipulation involving object interaction, friction, and constrained motion.

### A.2.1. CROPPED IMAGES

Fig. 7 shows the hardware setup for the motherboard assembly task, which presents the robot, the camera placements, and the task arrangement.

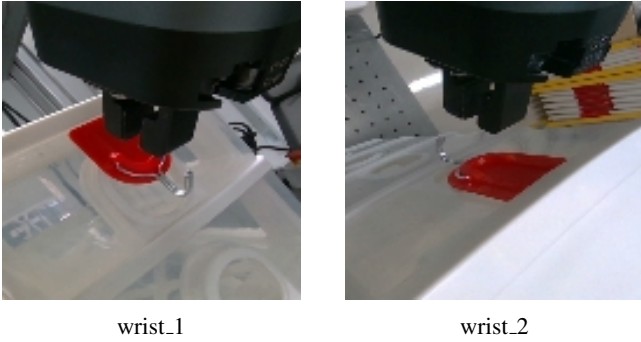

wrist_1        wrist_2

**Figure 7.** Sample input images from cameras used as inputs to the policy.

### A.2.2. POLICY TRAINING DETAILS

In Table 6, we report additional details of the policy training for this task.

**Table 6.** Policy training details for the Door Opening task.

| Parameter | Value |
| --- | --- |
| base policy | ACT |
| Observation space | wrist_1, wrist_2, tcp_pose, tcp_vel, tcp_f/t |
| Action space | 6D twist |
| Classifier views | wrist_2 |
| Classifier accuracy | 99% |
| Initial offline demonstrations | 30 |
| Max episode length | 100 environment steps |
| Reset method | Auto reset |
| Randomization range | 10 cm in x and y, 10 deg in rz |
| Proprio encoder size | 128 |
| Residual Policy MLP size | 256×256 |
| Discount factor | 0.98 |
| Optimizer | Adam |
| Image augmentation | Random crop + Random lighting intensity |
| xyz bound | 26cm*30cm*10cm |
| rpy bound | $[-50, 50]deg*0*0$ |

## A.3. Charger grasp-insertion

In the Charger Grasp-Insertion task, the robot first grasps a charging plug and then precisely inserts it into a corresponding socket on a power strip. The plug is initially placed at a nearby location with small variations in pose. Successful completion of the task requires accurate grasping, reliable pose alignment between the plug and the socket, and careful control during the insertion phase to avoid misalignment or collisions. This task emphasizes sequential manipulation, fine-grained spatial alignment, and contact-aware insertion under tight geometric constraints.

### A.3.1. CROPPED IMAGES

Fig. 8 shows the hardware setup for the motherboard assembly task, which presents the robot, the camera placements, and the task arrangement.

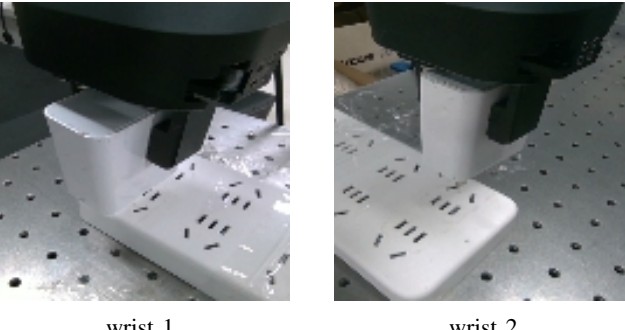

wrist_1                    wrist_2

**Figure 8.** Sample input images from cameras used as inputs to the policy.

### A.3.2. POLICY TRAINING DETAILS

In Table 6, we report additional details of the policy training for this task.

**Table 7.** Policy training details for the Charger grasp-insertion.

| Parameter | Value |
|---|---|
| base policy | ACT |
| Observation space | wrist_1, wrist_2, tcp_pose, tcp_vel, tcp_f/t |
| Action space | 6D twist + 1D gripper |
| Classifier views | wrist_2 |
| Classifier accuracy | 99% |
| Initial offline demonstrations | 30 |
| Max episode length | 100 environment steps |
| Reset method | Auto reset |
| Randomization range | 10 cm in x and y, 30 deg in rz |
| Proprio encoder size | 128 |
| Residual Policy MLP size | 256×256 |
| Discount factor | 0.98 |
| Optimizer | Adam |
| Image augmentation | Random crop + Random lighting intensity |
| xyz bound | 16cm*15cm*25cm |
| rpy bound | $0*0*[-60, 60]deg$ |

## A.4. Keychain hanging

In the Keychain Hanging task, the robot starts with the keychain ring already held in the gripper and must hang the ring onto a fixed hook. The initial pose of the ring relative to the hook is randomized within a small range. Successful completion requires precise alignment between the ring and the hook and careful, contact-aware motion to guide the ring over the hook and release it such that it remains securely suspended. This task emphasizes fine-grained pose control and contact-rich interaction under tight geometric constraints.

### A.4.1. CROPPED IMAGES

Fig. 9 shows the hardware setup for the motherboard assembly task, which presents the robot, the camera placements, and the task arrangement.

### A.4.2. POLICY TRAINING DETAILS

In Table 8, we report additional details of the policy training for this task.

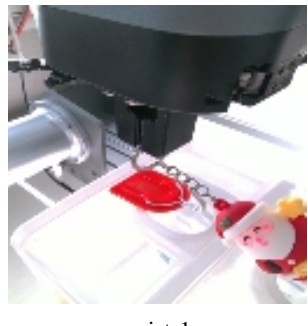 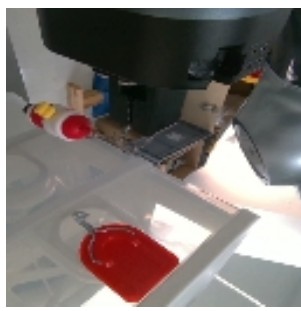

wrist_1                                wrist_2

**Figure 9.** Sample input images from cameras used as inputs to the policy.

**Table 8.** Policy training details for the Keychain hanging task.

| Parameter | Value |
|---|---|
| base policy | ACT |
| Observation space | wrist_1, wrist_2, tcp_pose, tcp_vel, tcp_f/t |
| Action space | 6D twist |
| Classifier views | wrist_2 |
| Classifier accuracy | 99% |
| Initial offline demonstrations | 30 |
| Max episode length | 100 environment steps |
| Reset method | Auto reset |
| Randomization range | 10 cm in x and y, 10 deg in rz |
| Proprio encoder size | 128 |
| Residual Policy MLP size | 256×256 |
| Discount factor | 0.98 |
| Optimizer | Adam |
| Image augmentation | Random crop + Random lighting intensity |
| xyz bound | 26cm*30cm*10cm |
| rpy bound | $[-10, 90]deg$*0*0 |

## A.5. USB Insertion (easy)

In the USB Insertion (Easy) task, the robot starts with a USB plug already held in the gripper and inserts it into a USB port on a charging adapter. The charging adapter is fixed in position and orientation throughout the task. Successful completion requires accurate alignment between the USB plug and the port and careful control during the insertion motion to avoid misalignment or collisions with the port edges. This variant primarily evaluates precise insertion control under constrained geometry with a known target pose.

### A.5.1. CROPPED IMAGES

Fig. 10 shows the hardware setup for the motherboard assembly task, which presents the robot, the camera placements, and the task arrangement.

### A.5.2. POLICY TRAINING DETAILS

In Table 9, we report additional details of the policy training for this task.

## A.6. USB Insertion (hard)

In the USB Insertion (Hard) task, the robot starts with a USB plug already held in the gripper and inserts it into a USB port on a charging adapter whose position is randomly selected from a set of predefined locations. The target port pose therefore varies across episodes. Successful completion requires the policy to first localize the target port and then perform accurate pose alignment and contact-aware insertion. Compared to the easy variant, this task introduces target pose uncertainty and emphasizes robustness to spatial variation in addition to precise insertion control.

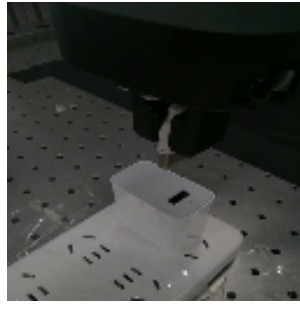 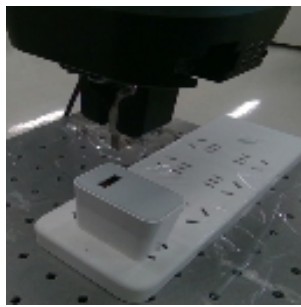

wrist_1          wrist_2

**Figure 10.** Sample input images from cameras used as inputs to the policy.

**Table 9.** Policy training details for the USB Insertion (easy) task.

| Parameter | Value |
|---|---|
| base policy | ACT |
| Observation space | wrist_1, wrist_2, tcp_pose, tcp_vel, tcp_f/t |
| Action space | 6D twist |
| Classifier views | wrist_1 |
| Classifier accuracy | 99% |
| Initial offline demonstrations | 30 |
| Max episode length | 100 environment steps |
| Reset method | Auto reset |
| Randomization range | 10 cm in x and y, 30 deg in rz |
| Proprio encoder size | 128 |
| Residual Policy MLP size | $256 \times 256$ |
| Discount factor | 0.98 |
| Optimizer | Adam |
| Image augmentation | Random crop + Random lighting intensity |
| xyz bound | 10cm*10cm*25cm |
| rpy bound | $0*0*[-60, 60]deg$ |

### A.6.1. CROPPED IMAGES

Fig. 11 shows the hardware setup for the motherboard assembly task, which presents the robot, the camera placements, and the task arrangement.

### A.6.2. POLICY TRAINING DETAILS

In Table 10, we report additional details of the policy training for this task.

## B. RL Module Design

Following the dual-buffer mechanism adopted in the RLPD (Ball et al., 2023), which consists of an RL replay buffer $\mathcal{D}_{online}$ and an expert demonstration buffer $\mathcal{D}_{offline}$, We sample half of the current batch size from each buffer every time the actor and critic are updated. Additionally, we adopt the delayed actor update approach from TD3, where actor updates occur every 2-4 critic updates, in order to reduce instability caused by updates with inaccurate value estimates. To further address value overestimation, we implement Clipped Double Q-Learning, which employs an ensemble of at least two Q-functions to compute the TD error. In line with HIL-SERL (Luo et al., 2025), we adopt a DrQ-style approach, utilizing a pretrained ResNet as the visual encoder, followed by a 2–3 layer MLP that serves as the network architecture for both the actor and the critic. For data augmentation, we apply random cropping and random adjustments to the lighting intensity.

## C. OOD Experiment Setting

## D. Visual Disturbances Experiment Setting

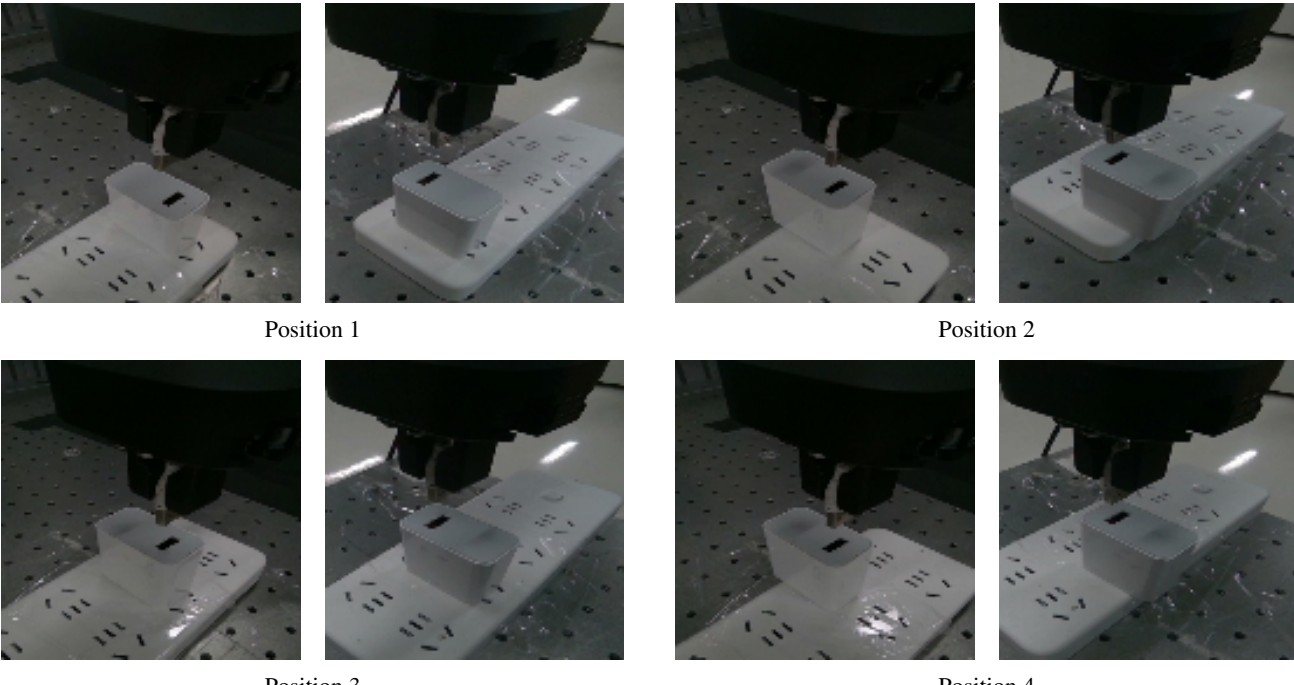

Figure 11. Sample input images from cameras used as inputs to the policy.

Table 10. Policy training details for the USB Insertion (hard) task.

| Parameter | Value |
| --- | --- |
| base policy | ACT |
| Observation space | wrist_1, wrist_2, tcp_pose, tcp_vel, tcp_f/t |
| Action space | 6D twist |
| Classifier views | wrist_1 |
| Classifier accuracy | 99% |
| Initial offline demonstrations | 30 |
| Max episode length | 100 environment steps |
| Reset method | Auto reset |
| Randomization range | 10 cm in x and y, 30 deg in rz |
| Proprio encoder size | 128 |
| Residual Policy MLP size | 256×256 |
| Discount factor | 0.98 |
| Optimizer | Adam |
| Image augmentation | Random crop + Random lighting intensity |
| xyz bound | 10cm*10cm*25cm |
| rpy bound | $0*0*[-60, 60]deg$ |

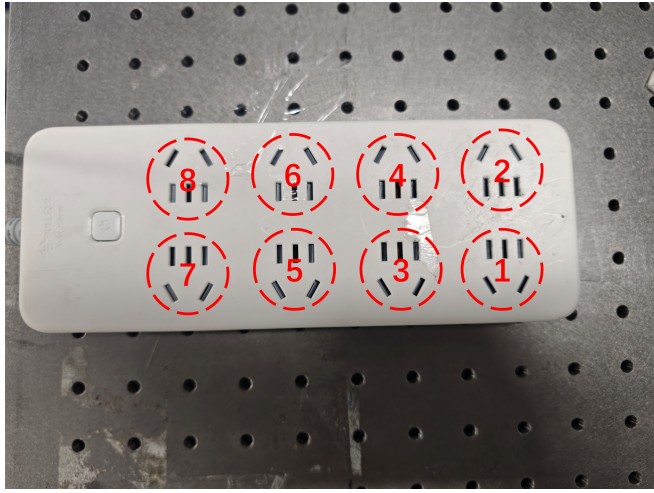

**Figure 12.** Our USB insertion task is trained at positions 1, 2, 3, and 4, with out-of-distribution (OOD) testing conducted at positions 5, 6, 7, and 8.

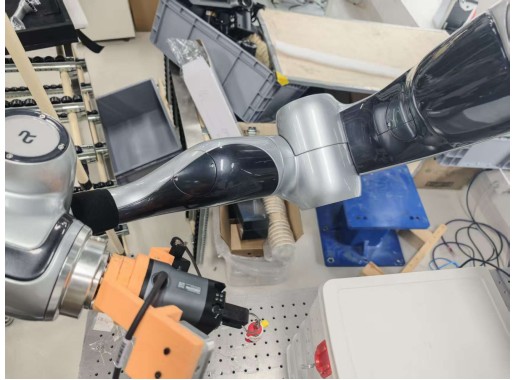
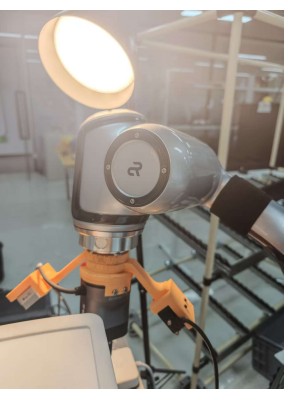
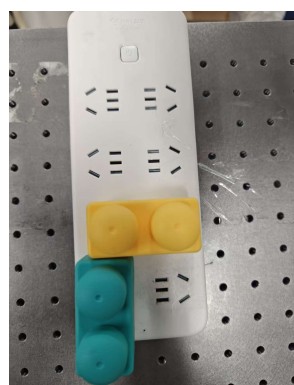

Ambient Changes                    Strong Lighting                    In-scene Occlusion

**Figure 13.** Visual disturbance experiment setting.

