# OpenReview forum: "Focus-Then-Contact: Speeding Up Robotic Contact-Rich Task Learning with Affordance-Guided Real-World Residual Reinforcement Learning"
_ICML.cc/2026/Conference — ICML 2026 regular_

### Official Review · Reviewer_h3RL · 2026-03-04

**Soundness:** 3
**Presentation:** 3
**Significance:** 3
**Originality:** 3
**Overall Recommendation:** 4
**Confidence:** 3

**Summary:**

- Proposes a Human-in-the-Loop Residual Reinforcement Learning approach that fine-tunes pre-trained Imitation Learning (IL) policies through online interaction.
- Uses Keyframe-based Affordance Guidance to provide dense visual rewards, directing the agent to "focus" on critical task regions before initiating high-precision "contact."
- Evaluations involve 6 contact rich tasks in the real world: insertion(card, charger, usb), hanging and door opening.

**Compliance With Llm Reviewing Policy:**

Affirmed.

**Final Justification:**

Based on rebuttal clarifications, I have raised my score.

**Key Questions For Authors:**

- Can you elaborate on the meaning of H in eq 4?
- How does eq3-4 capture the case when the human intervenes and the action is no longer related to the value of the policy and the base action.
- Even if the current ROI image depicts heavy occlusions (e.g usb port occluded by usb stick), how to  differentiate contact states just from visual embedding distance? For instance, if the ROI image seems to show usb stick in port, but it’s jammed.
- What is the difference in performance between BC, ACT and FCT in-domain scenarios? This to get an idea how well each method is capturing the data distribution?
- The table 1 results are when testing all policies frozen and not allowing extra human interventions?
- From table 2, is it necessary the residual policy with dense rewards? Seems like the human interventions is all that is needed for getting high success rates

**Limitations:**

Yes

**Strengths And Weaknesses:**

**Strengths**
- Residual policy informed by distance to goal state instead of vanilla sparse reward.
- Solution exploits the structure of the problem.
- The solution is technically sound and experiments support the possible improvements  to BC

**Weaknesses**
- The paper does not to quantify the specific reduction in human intervention frequency resulting from the residual policy and affordance-guided rewards. While overall training time is tracked, there is no direct measurement of how many fewer manual corrections are required to reach peak performance compared to standard baselines.
- The distance function based on InfoNCE for dense rewards although interesting seems limited compared to pretrained visual embeddings. Also unsure in general how this approach can handle differentiating visually similar  states but different in contact state.
- Mathematically, rotations (Euler) cannot be directly added. While the authors state this did not significantly impact convergence in their specific tasks, it remains a theoretical and algorithmic flaw that could lead to instability in more complex, high-degree-of-freedom rotational tasks
- If the initial demonstrations do not provide a clear view of the goal state, or if the ROI is poorly defined, the dense reward signal becomes unreliable, hindering the agent's ability to "focus" on the correct area
- The method still requires a human to be physically present to provide interventions, which limits its ability to scale to massive, unattended training regimes

---

> ### Author Rebuttal · Authors · 2026-03-31
>
> Dear Reviewer, we sincerely value your time and effort in evaluating our work. Your insights have been valuable to our work. We have prepared comprehensive responses and clarifications to address each point you raised. We hope these responses can resolve your concerns.
>
>
> 1. *"Can you elaborate on the meaning of H in eq 4?"*
>
> **Our response.**
> There is indeed a typo in Eq. 4; it should be $\mathcal{H}$, representing the entropy of the policy.
>
> 2. *" How does eq3-4 capture the case when the human intervenes, and the action is no longer related to the value of the policy and the base action. "*
>
> **Our response.** Our human intervention mechanism works as follows:
> When intervention occurs, the actual action executed $a_{human}$ is provided directly by the human, rather than $\pi_{base}+\pi_{res}$.
> The trajectories generated by these interventions are stored separately into two buffers:
>
> - $D_{online}$: The online replay buffer for RL, used for SAC training. It consists of human interventions and RL rollout trajectories.
> - $D_{offline}$: The expert demonstration buffer. It consists of human interventions and expert trajectories.
>
> During training, the RL algorithm samples 50% of the data from $D_{online}$ and 50% from $D_{offline}$, which may contain transitions (state, human action, reward, next state) produced by human interventions as well as expert trajectories. Since the Q-function and policy are trained on this mixed data, they indirectly learn the *correction signal* brought by human interventions.
>
> 3. *"Even if the current ROI image depicts heavy occlusions (e.g usb port occluded by usb stick), how to differentiate contact states just from visual embedding distance? For instance, if the ROI image seems to show usb stick in port, but it’s jammed."*
>
> **Our response.** We visualize the distance function corresponding to the trajectories during the rollout of RL in https://anonymous.4open.science/api/repo/rebuttal-5B45/file/review.html. Even if the current ROI image depicts heavy occlusions, the distance function can still reflect the trend of reward changes when the gripper is occluded or stuck on the hole. For example, when occlusion occurs during USB insertion, the value of the distance function immediately increases, and after realigning with the hole, the distance decreases.
>
> 4.*"What is the difference in performance between BC, ACT and FCT in-domain scenarios? This to get an idea how well each method is capturing the data distribution? "*
>
> **Our response.** For example, in tasks such as USB insertion, charger insertion, and cart insertion, BC (MLP) can descend but leaves a relatively large gap to the target position. ACT can get closer to the target position, but sometimes, as in USB insertion, it fails to remain on the surface of the charger and instead slides off the side. In contrast, FTC successfully learns the appropriate angle and force to accurately insert into the target position.
>
> 5.*"The table 1 results are when testing all policies frozen and not allowing extra human interventions?"*
>
> **Our response.** Your understanding is accurate. The table 1 results are when testing all policies frozen and not allowing extra human interventions.
>
> 6.*"From table 2, is it necessary the residual policy with dense rewards? Seems like the human interventions is all that is needed for getting high success rates"*
>
> **Our response.** We think that when the task has a certain level of difficulty and generalization requirements, it is still necessary to incorporate dense rewards into the residual policy. Human-in-the-loop indeed contributes significantly to achieving success rates, but our goal is to minimize training time per task and reduce the burden on the human operator. During our experiments in Table 2 of the paper, we found that the residual policy and dense rewards complement each other, both contributing to shorter training times for real-world RL. Even with human intervention to correct RL's mistakes, most of the corrected trajectories still receive a reward signal of zero, which provides limited guidance to the residual policy. The videos in the https://anonymous.4open.science/api/repo/rebuttal-5B45/file/review.html illustrate that during contact between the RL agent and the object, dense rewards can reflect the gap between the current state and task completion. We have also recorded the number of human interventions in https://anonymous.4open.science/api/repo/rebuttal-5B45/file/review.html, and the results display that compared to sparse rewards, dense rewards based on the residual policy can reduce the frequency of human interventions.

---

> > ### Author Rebuttal · Reviewer_h3RL · 2026-04-01
> >
> > Thanks for the responses. Given the clarifications and the shared supplementary material I have more evidence that the method is in the direction of reducing the amount of human interventions. I have updated my rating accordingly.

---

> > > ### Author Response · Authors · 2026-04-02
> > >
> > > Dear reviewer, thank you very much for your positive feedback and the valuable suggestions you provided during the rebuttal period. We have learned a lot from them and have revised the final version of the paper based on your comments. Thank you!

---

### Official Review · Reviewer_ckYS · 2026-03-12

**Soundness:** 2
**Presentation:** 3
**Significance:** 2
**Originality:** 3
**Overall Recommendation:** 4
**Confidence:** 4

**Summary:**

This paper proposes a human-in-the-loop real-world reinforcement learning framework designed to accelerate policy learning for contact-rich robotic manipulation tasks. The framework learns a lightweight residual RL policy on top of a pretrained imitation learning base policy, with human intervention available during training to guide exploration. To further accelerate learning, the authors introduce a keyframe-based affordance reward that measures the visual embedding distance between the current observation and a goal frame extracted from offline demonstrations. The framework also includes a carefully designed human intervention window to prevent conflicts between human demonstrations and RL control during training. Experiments across six contact-rich tasks show that FTC achieves 100% success rates with reduced training time compared to the baselines, and generalizes reasonably well to certain out-of-distribution setting.

**Compliance With Llm Reviewing Policy:**

Affirmed.

**Final Justification:**

Based on the overall discussions, I decided to keep my score unchanged.

**Key Questions For Authors:**

Questions:
- Does the training time in Table 1 include both data collection and model update time? It would also be helpful to report the number of online trajectories to clearly decouple data efficiency from other sources of wall-clock time.
- Is a scaling factor applied to the residual action? If so, do the authors have practical recommendations for tuning it?
- Line 272 says "Figure 11 illustrates that the object detection in SAM 3 can be affected by the surrounding environment", but Figure 11 appears unrelated to this claim.

**Limitations:**

yes

**Strengths And Weaknesses:**

Strengths:
- The paper is well-written and easy to follow.
- The proposed method performs well compared against the selected baselines, especially on more challenging tasks and OOD settings.
- The accompanying video materials demonstrate performance across different initial configurations & OOD cases, which strengthens the reliability of the experimental results.

Weakness: Among the key components of FTC, human-in-the-loop learning and residual RL are both well-established practices in the existing literature, leaving the keyframe-based affordance reward as the primary novel contribution. However, this component has some notable limitations. The reward implicitly assumes that the visual embedding distance to a fixed goal keyframe within a predefined ROI provides meaningful guidance throughout the entire task trajectory. This makes it less applicable to other contact-rich tasks where meaningful policy improvement is needed well before the robot reaches the target region. For instance, in tasks like pushing an object to a target location or cloth folding, the keyframe-based reward would provide little useful signal for the majority of the interaction.

---

> ### Author Rebuttal · Authors · 2026-03-31
>
> Dear Reviewer, we sincerely value your time and effort in evaluating our work. Your insights have been valuable to our work. We have prepared comprehensive responses and clarifications to address each point you raised. We hope these responses can resolve your concerns.
>
>
> 1. *"Does the training time in Table 1 include both data collection and model update time? It would also be helpful to report the number of online trajectories to clearly decouple data efficiency from other sources of wall-clock time."*
>
> **Our Response**. The training time reported in Table 1 does not include the time spent collecting offline expert trajectories, and only accounts for the interaction time of collecting online trajectories and the model update time. We adopt a client-server architecture for the actor and critic, allowing online data collection and model updates to be performed synchronously. Table 1 shows the number of online trajectories collected for real-world RL. We can see that even though HIL-SERL uses a longer training time than FTC, it does not obtain many online trajectories for updating the actor and critic. This is because HIL-SERL utilizes more human intervention steps and intervention counts to correct RL's mistakes, which slows down the rate of collecting online trajectories. For example, in the charger insertion task, FTC begins to achieve some success as early as the 12th minute, which speeds up the collection of online trajectories.
>
> **Table 1: The number of online trajectories**
>
> | Method       | Card insertion | Door opening | Charger grasp-insertion | Keychain hanging | USB insertion (easy) | USB insertion (hard) |
> |--------------|----------------|--------------|-------------------------|------------------|----------------------|----------------------|
> | HIL-SERL     | 66             | 31           | 50                      | 39               | 139                  | ≥168                 |
> | FTC          | 50             | 26           | 47                      | 30               | 54                   | 126                  |
>
>
> 2. *"Is a scaling factor applied to the residual action? If so, do the authors have practical recommendations for tuning it? "*
>
> **Our response.** We apply a scaling factor. Based on our experience, you can adjust this parameter according to the performance of your base policy. Since our current method for collecting teleoperation data for the robotic arm relies solely on a SpaceMouse, data quality inevitably affects the performance of the base policy. If the base policy performs well, we think the coefficient can be appropriately reduced; if performance is moderate, it can be increased accordingly.
>
>
> 3. *"Line 272 says "Figure 11 illustrates that the object detection in SAM 3 can be affected by the surrounding environment", but Figure 11 appears unrelated to this claim."*
>
> **Our response.** Thank you for your careful reading of our manuscript. It should be Figure 2. We apologize for any confusion caused and appreciate the valuable suggestions.

---

> > ### Author Rebuttal · Reviewer_ckYS · 2026-04-01
> >
> > Thank you for the clarifications. The issues raised in the Questions part have been fully resolved. In subsequent revisions, I encourage the authors to discuss the concerns noted in my Weakness part, along with potential directions for future improvement. My rating remains unchanged.

---

> > > ### Author Response · Authors · 2026-04-02
> > >
> > > Dear reviewers, Thank you very much for your encouraging feedback and valuable suggestions. We train our keyframe-based visual embedding distance reward on more atomic tasks involving contact-rich insertion and extraction. Based on the examples you provided, we conduct a detailed discussion of this method's limitations and future directions:
> > >
> > > **Discussion of Limitations**:
> > >
> > > *Non-monotonic Visual Changes*:
> > > Obstacle-avoidance pushing: During the process of pushing an object to a target location, the relative pose between the robot and the object, as well as the object's orientation, continuously changes. Visually, the embedding distance between the current frame and the keyframe of "object at target position" is not monotonically decreasing. To navigate around obstacles or adjust the pushing direction, the policy may need to temporarily move the object away from the target, increasing the visual distance, but this is necessary for the correct strategy. Distance-based rewards would incorrectly penalize such essential behaviors.
> > >
> > > *Process-oriented Tasks*:
> > > Cloth folding: Successful folding does not depend on reaching a specific "keyframe" configuration of the final cloth state, but rather on executing a correct sequence of operations (e.g., lifting one corner, moving it to the opposite side, placing it down, smoothing). During most intermediate steps, the embedding distance between the current visual state and the final "fully folded" state is large and meaningless. The reward function cannot inform the policy that "lifting this corner" is a good intermediate action, as it does not bring the state any closer to the final keyframe.
> > >
> > > *Single-view Limitation*：
> > > Due to the single-view limitation of keyframe-based rewards, a single viewpoint may already be sufficient for atomic tasks such as inserting. However, if we want to extend to long-horizon tasks such as cooking, one viewpoint is often not enough. A third-person perspective camera is needed to simultaneously record the entire process of the task, so that the trained reward model can provide clear guidance for the task.
> > >
> > > **Future Directions**:
> > >
> > > In the future, we think reward models should incorporate step-aware understanding and rely on multi-view perception, leading to reliable assessments of contact-rich manipulation progress. Moving beyond keyframe-based outcome verification, we think a process reward model should explicitly track which sub-goals have been achieved, validate the physical plausibility of state transitions, and forecast the feasibility of remaining sub-goals. The resulting dense rewards can provide continuous reward signals at every timestep, or discrete rewards upon completion of each sub-step, with a large final reward when the task is fully completed. Additionally, this reward model should possess visual robustness, and it should focus only on the robotic arm and task-relevant objects, ignoring surrounding objects as well as variations in lighting and color temperature, thereby better enhancing the generalization of real-world RL deployment.
> > >
> > > We can explore two approaches: 1. Leveraging the visual capabilities of VLMs to collect a large number of expert demonstrations for fine-tuning, using the task completion status as a reference. Data sources can include human operation videos from the internet, expert trajectories obtained via teleoperation, and data generated through motion planning/agents in simulation. 2. Inspired by the RLHF paradigm of large language models, we can generate preference supervision by comparing complete expert trajectories and use this as an auxiliary signal to learn a generalizable reward function that works across different tasks and embodiments.
> > >
> > > We have discussed the concerns noted in your Weakness part, along with potential directions for future improvement, and refined the final version of the paper based on your feedback. Thank you!

---

### Official Review · Reviewer_GKBe · 2026-03-13

**Soundness:** 3
**Presentation:** 3
**Significance:** 3
**Originality:** 2
**Overall Recommendation:** 4
**Confidence:** 5

**Summary:**

This paper proposes **Focus-Then-Contact (FTC)**, a real-world learning framework for contact-rich insertion tasks. The approach integrates pretraining and human-in-the-loop (HIL) data collection with online reinforcement learning. In addition, a latent-distance-based reward is introduced to accelerate learning. The authors validate their method through extensive real-world experiments.

**Compliance With Llm Reviewing Policy:**

Affirmed.

**Key Questions For Authors:**

1. Regarding evaluation: What measures were taken to **ensure a fair comparison between baselines**, given the human factors involved in data collection?

2. Could the authors provide more insight into the **failure modes of HIL-SERL**, especially in the USB-hard task? Understanding these failure patterns would clarify the advantages of FTC.

**Limitations:**

Yes

**Strengths And Weaknesses:**

## Strengths
- **Challenging problem:** Contact-rich insertion tasks are notoriously difficult. The authors demonstrate a USB-hard insertion task where a prior method, HIL-SERL, fails.
- **Clear presentation:** The paper is well-written, structured, and easy to follow.
- **Strong engineering effort:** The experimental setup and real-world validation show careful attention to implementation details.

## Weaknesses
- **Integration of existing components:** The method primarily combines existing learning modules. Residual policies, mixed training strategies, and distance-based reward functions have all been studied in prior works in separate contexts. As a result, the main contribution is system integration, and the scientific novelty appears limited.
- **Human factors and evaluation challenges:** The involvement of human-in-the-loop data collection introduces variability, making fair comparisons difficult. There is limited discussion on the failure patterns of the strongest baseline, which is important to understand the relative performance of FTC.

---

> ### Author Rebuttal · Authors · 2026-03-31
>
> Dear Reviewer, we sincerely value the time and effort you have devoted to evaluating our work. To address each point you raised, we have prepared comprehensive responses and clarifications. We hope these responses can resolve your concerns.
>
> 1. *Regarding evaluation: What measures were taken to ensure a fair comparison between baselines, given the human factors involved in data collection?*
>
> **Our response.** To ensure a fair comparison between baselines, we made the following decisions:
>
> (1) For BC (MLP), we use 50–100 trajectories with the checkpoint from epoch 20,000, and for ACT we choose the checkpoint from epoch 7000. Our base policy uses 40 trajectories with the checkpoint from epoch 4,000.
>
> (2) For HIL-SERL, HGDagger, and ResFit, the actor is warm-started, while FTC is not.
>
> (3) HIL-SERL, HGDagger, and ResFit use 40–50 expert trajectories to initially fill the offline buffer, while FTC uses 30 trajectories.
>
> (4) All methods are evaluated under the same task setting with random starting and ending positions and apply the same data augmentation and impedance coefficients.
>
> (5) During testing, the surrounding environment is kept unchanged to eliminate interference caused by environmental variations.
>
> (6) All other hyperparameters, such as discount factor, batch size, UTD ratio, and max steps, are kept the same across methods.
>
> (7) All offline datasets are collected using 3D spacemouse teleoperation.
>
> (8) All baselines are trained with the same type of GPU.
>
> 2. *Could the authors provide more insight into the failure modes of HIL-SERL, especially in the USB-hard task? Understanding these failure patterns would clarify the advantages of FTC.*
>
> **Our response.** We think the main reason HIL-SERL fails in the USB insertion hard task is insufficient training time and not fully applying RL where it works best. If there were no time constraints, we think HIL-SERL could still accomplish this task, but it would require a lot of time and place a heavy burden on the human operator. Insertion-type tasks generally consist of two stages: the first is descending to the vicinity of the target position, and the second is making contact adjustments to enable insertion. The main advantage of RL should be applied as much as possible to the second stage. In the USB insertion hard task, HIL-SERL spends a great deal of learning time just on the first stage, descending to the ROI. Often, HIL-SERL does not spend time staying at the surface of the charger to learn the skill through prolonged contact and repeated insertion attempts. The advantage of FTC is that it uses a base policy and an affordance-based reward to accelerate the first stage, allowing the robotic arm sufficient time to continuously make contact and perform insertions near the target position, so the residual policy quickly learns this atomic skill.

---

> > ### Author Rebuttal · Reviewer_GKBe · 2026-04-02
> >
> > Thank you for the update!

---

> > > ### Author Response · Authors · 2026-04-02
> > >
> > > Dear reviewers, You are welcome! Thank you very much for your encouraging feedback and valuable suggestions. We have refined the final version of the paper based on your feedback. Thank you!

---

### Official Review · Reviewer_cQV5 · 2026-03-13

**Soundness:** 2
**Presentation:** 3
**Significance:** 3
**Originality:** 2
**Overall Recommendation:** 4
**Confidence:** 3

**Summary:**

This paper proposes Focus-Then-Contact (FTC), a human-in-the-loop real-world RL framework for contact-rich manipulation like card insertion, charger insertion, keychain hanging, and USB insertion. The core claim is that real-world RL for these tasks is slow because exploration is expensive, sparse rewards are uninformative, and humans must intervene too much. FTC tries to reduce that burden by combining three pieces: a frozen imitation-learned base policy, a lightweight residual RL policy, and a dense affordance-style reward built from a goal keyframe.

**Compliance With Llm Reviewing Policy:**

Affirmed.

**Final Justification:**

I am satisfied with the reviewer rebuttal enough to raise my score to weak accept.

**Key Questions For Authors:**

1. **On reward scalability.** The authors note that the reward model requires separate data collection and training for each task, and that long-horizon settings may need a more general reward model. How much per-task supervision is required in practice, and do you have any evidence that the keyframe-based reward could transfer across related tasks, objects, or stages of a longer-horizon manipulation problem?

2. **On robustness/generalization scope.** The paper claims robustness and generalization, but the direct evidence seems fairly narrow: the OOD study is centered on USB insertion and Table 10 shows severe degradation under strong lighting. Could you narrow the robustness claims, or add broader evaluations across more tasks, disturbance types, and unseen object/scene configurations? Some discussion on what specifically breaks under strong lighting and occlusion cases could also be helpful; for example, is the main failure mode in the visual encoder, the ROI localization, the success classifier, or the policy’s contact behavior once near the goal?

**Limitations:**

yes

**Strengths And Weaknesses:**

Strengths
- **Technically sound overall.** The main empirical claims around efficiency and accuracy are well-supported by the experiments. FTC is evaluated on 6 real-world contact-rich tasks, and beats the listed baselines in success rate and training speed. The ablations show that the residual policy and affordance-guided reward help performance, especially on the harder USB setting.
- **Meaningful practical significance.** The paper targets an important problem setting in robot learning, aiming to achieve sample-efficient real-world learning for contact-rich manipulation, where sparse rewards and heavy human intervention are costly. A method that speeds up convergence while improving robustness and OOD position generalization would be valuable for real robotic practice.

Weaknesses
- **Robustness and generalization claims are a bit broader than the evidence supports.** The paper emphasizes robustness and generalization, but the evidence is concentrated on a limited set of tasks with the authors acknowledging drastic performance drops under strong lighting changes and occlusions (e.g. 0% success rate for the Charger grasp-insertion task under strong lighting changes in Table 10). So FTC's performance and efficiency gains over baselines are solid, but the broader robustness claims should be stated somewhat more narrowly.
- **Reward design has scalability limitations.** The authors acknowledge that the reward design is task-specific, requiring separate
data collection and training for each task, which may limit extension to more diverse or long-horizon tasks. The authors additionally acknowledge unexplored trade-offs between reward model size/inference speed and reward generalization.
- **Limited methodological novelty.** The paper’s main contribution is a practical combination of existing methods and ideas rather than a clearly new algorithmic or theoretical insight.

---

> ### Author Rebuttal · Authors · 2026-03-31
>
> Dear Reviewer, we sincerely value the time and effort you have devoted to evaluating our work. To address each point you raised, we hope these responses can resolve your concerns.
>
> 1. *"On reward scalability. The authors note that the reward model requires separate data collection and training for each task, and that long-horizon settings may need a more general reward model. How much per-task supervision is required in practice, and do you have any evidence that the keyframe-based reward could transfer across related tasks, objects, or stages of a longer-horizon manipulation problem?"*
>
> **Our Response.** We discuss dense reward and sparse reward separately. For sparse reward, we train a separate binary reward classifier for each task. We need to collect four successful trajectories and eight failed trajectories per task, with each trajectory containing 50 to 100 images. Training and inference are done on a single 4090 GPU for 2,000 epochs, but the total time does not exceed five minutes. Therefore, even training a sparse reward model individually for each task is very fast.
>
> For dense reward, we use a shared visual encoder for all tasks. To ensure that the visual encoder generalizes to most tasks, we first pretrain it on a large amount of human manipulation videos. Although out-of-domain videos are not in-domain data for robot control, they can be considered in-domain data for human action strategies. The link https://anonymous.4open.science/api/repo/rebuttal-5B45/file/review.html shows the dense reward curves for various tasks under the same visual encoder, indicating that the keyframe-based reward can transfer across related tasks and objects. The first row of animations clearly illustrates the fluctuations during different contact-rich tasks. In contrast, the second row distinctly illustrates the two-step process of pick-and-place tasks. Additionally, although the objects differ between the left and right figures, the reward trends remain largely consistent.
>
> 2. *"On robustness/generalization scope. The paper claims robustness and generalization, but the direct evidence seems fairly narrow: the OOD study focuses on USB insertion, and Table 10 presents severe degradation under strong lighting. Could you narrow the robustness claims, or add broader evaluations across more tasks, disturbance types, and unseen object/scene configurations? Some discussion on what specifically breaks under strong lighting and occlusion cases could also be helpful; for example, is the main failure mode in the visual encoder, the ROI localization, the success classifier, or the policy’s contact behavior once near the goal?"*
>
> **Our Response.**  We thank the reviewer for providing us with the methods for reference and comparison. Through experiments, we conduct a more rigorous scope of generalization/robustness, including:
>
> - Task A: variations of objects around the workspace
> - Task B: placing tablecloths of different colors
> - Task C: OOD objects appearing within the ROI region
> - Task D: waving hands in front of the camera during execution
> - Task E: Strong light exposure
> - Task F: Exposure to different colored lights
> - Task G: External forces pushing the robotic arm
> - Task H: Turn the indoor lights on or off
> - Task I: Slightly adjust the insertion position within a small range during execution
> - Task J: Replace with a USB plug/cable of a different color
>
> We evaluate each task 10 times and record the success rate under USB insertion (easy) and charger insertion tasks. We redefine the scope of generalization. For tasks where the method performs well (Tasks A, D, G, H, I) in Table 1, this indicates that the policy is robust at the motion execution level and against unstructured dynamic disturbances in time and space (e.g., object position shifts, hand waving, external forces, different natural lighting conditions). For tasks where the method performs poorly (Tasks B, C, F, J, E), this suggests that the visual perception layer is sensitive to variations in color, texture, lighting color, and lighting intensity.
>
> For the failed tasks, we think the primary cause of failure is the insufficient robustness of the visual encoder. Since the object colors in the training data are largely fixed, the feature maps produced under different colors or strong lighting conditions all affect the FTC. Additionally, different colors and strong lighting introduce errors in ROI localization. However, our main focus remains on improving the training efficiency of real-world RL and reducing the burden on human operators.
>
> **Table 1: Generalization scope**
>
> | Task                 | A    | B    | C    | D    | E     | F    | G    | H    | I    | J    |
> |----------------------|------|------|------|------|-------|------|------|------|------|------|
> | USB insertion (easy) | 100% | 0%   | 0%   | 100% | 20%   | 0%   | 100% | 100% | 100% | 0%   |
> | charger insertion    | 100% | 0%   | 0%   | 100% | 10%   | 0%   | 100% | 100% | 100% | 0%   |

---

> > ### Author Rebuttal · Reviewer_cQV5 · 2026-04-03
> >
> > Thank you for responding. Give me more time to think about this paper and I will update my score to reflect that soon

---

> > > ### Author Response · Authors · 2026-04-03
> > >
> > > Dear Reviewer, Thank you very much for your understanding and patience. Your willingness to give us more time to carefully consider the paper and your promise to update your score afterwards are truly invaluable to us. We have revised the paper based on your positive feedback and valuable comments. Thank you again for the time and professional review you have dedicated to this manuscript!

---

### Decision · Program_Chairs · 2026-04-30

**Decision:**

Accept (regular)

**Comment:**

The introduced method, Focus-Then-Contact, tackles the difficult problem of real-world contact-rich manipulation. The main takeaway is that integrating a pretrained imitation learning base policy with lightweight residual reinforcement learning and keyframe-based affordance rewards effectively increases sample efficiency. Reviewers appriciated the highly compressed rebuttal contributions which clarified evaluation fairness measures and detailed the specific HIL-SERL failure modes. The reviewers align on a weak accept rating without any score disagreements. This is a practical contribution that will benefit the conference.